# Pain Reduction in Patellofemoral Knee Patients During 3-Month Intervention with Biomechanical and Sensorimotor Foot Orthoses: A Randomized Controlled Clinical Study

**DOI:** 10.3390/biomedicines13010038

**Published:** 2024-12-27

**Authors:** Steven Simon, Andreas Heine, Jonas Dully, Carlo Dindorf, Oliver Ludwig, Michael Fröhlich, Stephan Becker

**Affiliations:** 1Department of Sports Science, RPTU University of Kaiserslautern-Landau, 67663 Kaiserslautern, Germany; jonas.dully@rptu.de (J.D.); carlo.dindorf@rptu.de (C.D.); oliver.ludwig@rptu.de (O.L.); michael.froehlich@rptu.de (M.F.); stephan.becker@rptu.de (S.B.); 2Orthopedic Medical Practice, Hammer Straße 9, 49740 Haselünne, Germany; heine@orthopaedie-emsland.de

**Keywords:** sensorimotor system, SMFO, foot orthoses, sensorimotor insoles, knee pain patients, chondropathia patellae, orthopedic treatment, Kujala Anterior Knee Pain Scale, visual analog scale

## Abstract

**Background:** Patellofemoral pain (PFP) significantly affects patients’ daily activities and consequently reduces their quality of life. Custom-made foot orthoses (FOs) are a common method of medical treatment that positively influences biomechanical factors such as the kinematics of the lower extremity and reduces pain perception in patients. However, there is a gap in research regarding the influence of different FO treatments on knee pain. Therefore, this study addresses the impact of biomechanical foot orthoses (BMFOs) and sensorimotor foot orthoses (SMFOs) on patients with foot deformity and PFP. **Methods:** A total of 26 participants (9 men, 17 women; 27.7 ± 10.7 years; 175.0 ± 0.1 cm; 75.7 ± 18.8 kg; BMI: 24.7 ± 5.6) took part in this randomized controlled clinical trial. In the pre-test, knee pain was evaluated using the Kujala Anterior Knee Pain Scale after the physician’s anamnesis and plantar pressure measurement. A 3-month intervention with SMFO and BMFO was performed, and weekly development was evaluated using 11-item visual analog scales (VASs). Repeated measures analyses of variance were used to assess differences between time of measurements (ToMs) and the interaction effect between ToMs and treatment groups (SMFO, BMFO). **Results:** Statistical analysis revealed no statistically significant interaction between ToMs and treatment groups but a significant main effect on Kujala anterior knee pain scores (M_Diff_ = 10.189; *p* = 0.014) and 12-week VAS (*p* = 0.001). **Conclusions:** The findings indicate that both treatment approaches effectively alleviated perceived knee pain in the PFP sample with foot deformity, with neither approach demonstrating superior efficacy. This trial was registered in the WHO International Clinical Trials Registry Platform (ICTRP) and German Clinical Trials Register (DRKS00035082).

## 1. Introduction

Patellofemoral pain (PFP) syndrome is one of the most common causes of anterior knee pain in adolescents and adults [1,2]. It affects 23% of the total population [3]. There is evidence that PFP significantly restricts the quality of life of those affected [4,5], particularly as it is discussed as an indicator of patellofemoral osteoarthritis [6]. These pathologies often place a substantial burden on the healthcare system [7,8]. The term chondropathia patellae is also frequently used, associated with impaired patellofemoral kinematics [9]. The causes of these pain symptoms are multifactorial; therefore, many biomechanically related etiological factors and treatments are linked to the pathology of PFP [1]. Static and dynamic components, including altered tibial and femoral kinematics [10], increased knee abduction, ligament injuries [11], and muscular overpowering due to increased muscle forces negatively affecting patella guidance [12], might be causative factors. It is unclear whether this influences the development of PFP [1,13]. Foot and ankle postures and excessive pronation of the foot are considered to influence the knee abduction moment in the frontal plane and the ground reaction forces [14,15]. A high pronation speed during running can lead to internal rotation of the tibia [16]. From a biomechanical perspective, the target is to redirect the forces acting on the knee. Hoglund et al. [17] found that PFP patients had an increased hip and pelvic range of motion during the step-down test in the frontal and transverse planes but a reduced or nearly equal range of motion for these variables during single-leg squats. Correction of increased rear foot eversion might reduce the internal rotation of the tibia and femur, thereby reducing pelvic anteversion [18]. In general, the influence of altered foot function and plantar pressure on PFP is variable and unclear [10].

Custom-made foot orthoses (FOs) are medical aids used to help patients with lower extremity pain [19], foot and ankle motion [20], and excessive foot pronation in gait and running. Depending on the cause and symptoms, FOs are customized to achieve more favorable kinematic (e.g., joint angle) and kinetic (e.g., force peaks) conditions to relieve lower extremity joint pain [21]. On the one hand, Saxena and Haddad [22] concluded that FOs are an effective treatment option for relieving clinical symptoms of PFP, especially in young people. Gross and Foxworth [23] stated that FOs have a positive impact on PFP patients with excessive foot pronation, lower extremity alignment, and increased Q-angle. A randomized clinical trial by Collins et al. [20] concluded that FOs improve perceived knee pain rated on a visual analog scale (VAS), exhibit better short-term improvements than flat inserts, and have a similar effect to physiotherapy. Barton et al. [24] provided limited evidence suggesting that prefabricated FOs may reduce transverse plane knee rotation and offer greater short-term relief than flat insoles for individuals with PFP syndrome. Lewinson et al. [25] investigated the potential of modifying the angular impulse magnitude of knee abduction through lateral and medial wedged FOs to alleviate pain in runners with PFP and found a clinically significant pain reduction. On the other hand, Kayll et al. [26] concluded that medial support insoles did not alter patellofemoral joint loads during walking and running.

In orthopedic care, it is common to prescribe FOs to influence knee pathologies. In general, there is a distinction between the two main FO approaches: biomechanical foot orthoses (BMFOs) and sensorimotor foot orthoses (SMFOs; synonymously SMIs) [27]. BMFO is characterized by supporting and bedding elements that are primarily intended to provide support, correction, and relief, comparable to the aforementioned approaches with a medial and lateral wedge. In contrast, SMFO primarily aims to influence the activity of defined muscles via corresponding elements (e.g., medial or lateral hindfoot elements, toe bridges, and retrocapital elements) at a specific time interval in the step cycle in a targeted manner (tonizing and detonizing) [28,29]. The key therapeutic target of SMIs is to activate the Musculus (M.) peroneus longus and M. tibialis posterior, which stabilize the ankle joint. This stabilization improves foot positioning and pressure distribution, potentially reducing foot pain. In consequence, the adjustment in joint kinematics might alleviate discomfort in other joint segments due to biomechanical interconnections [27]. Kerkhoff et al. [30,31] examined prefabricated BMFOs and SMFOs and their effects on lower extremity muscle activity in participants with non-specific knee pain. The data showed that the prefabricated BMFO and custom-made SMFO led to different activation patterns compared with a shoe without FOs during a single-leg landing test, whereas SMFOs led to an increased musculus semitendinosus and M. peroneus longus influence. Chondropathia patellae is listed by Greitemann et al. [29] on the German DGOOC advisory committee as an indication for SMFO; however, there is a lack of randomized controlled clinical trials investigating whether FOs, especially SMFOs, have a positive effect on knee pain and which patients benefit most from FOs [24]. Therefore, this study targeted to address this research gap.

This study aimed to investigate the following research question: Do SMFOs reduce the perception of knee pain in PFP patients with foot deformity over a 3-month intervention period, observing between- and within-FO-group differences?

## 2. Materials and Methods

### 2.1. Study Design

This was a stratified randomized controlled clinical trial with pre- and post-testing. The intervention period was 3 months. The sample was randomly assigned by the test supervisor to an orthopedic custom-made device (SMFO, BMFO) over the intervention period after diagnostic and orthopedic anamnesis by the physician, considering the inclusion and exclusion criteria (Table 1 and Section 2.2) and stratified according to the localization of knee pain (anterior, retro patellar). The intervention group was treated with SMFO, and the control group was treated with BMFO. This trial was blinded, which means that the participants were not informed about their assigned intervention. The health provider was also not informed about this fact. The study was conducted from March to November 2024.

Scientific evaluation was performed without disrupting the standard procedure of the physician and orthopedic technician who treated patients with corresponding orthopedic indications. The study was conducted in accordance with the Declaration of Helsinki and approved by the ethics committee (No. 70, 16 February 2024) and registered in the German Clinical Trials Register (DRKS00035082; 17 September 2024).

A sample size of 24 participants was determined for an analysis of variance (ANOVA) to assess the interaction effect (effect size f = 0.25, 2 groups, 2 ToMs, α error probability = 0.05, correlation among repeated measures: 0.8) using G*Power 3.1 [32].

### 2.2. Sample

Participants were recruited from everyday patient care provided by the responsible physician. Anthropometric data are shown in Table 1.

The following inclusion and exclusion criteria were defined, and the same physician was responsible for the medical assessment of patients.

Inclusion criteria:Age between 15 and 60 yearsDiscomfort in the knee joint area during at least two weight-bearing activities (walking stairs, squatting, standing up) for at least 3 weeks: pain during these activities on most days in the last month that is ≥30 mm on a 100 mm VASIndication (at least one diagnosis from the following list):
oFemoropatellar pain syndromeoChondropathia patellae up to grade 3 with pathological alignment and femoral antetorsionoRunner’s knee, jumper’s kneeoOsteochondral defects, inflammation, and impingement of the Hoffa fat bodyoTendinopathies of the patellar or quadriceps tendon, patellofemoral osteoarthritis, plica syndromeAltered Q-angle [33] of the lower extremity/recognizable rotational abnormality of ankle joints, tibia, and femur during gaitFoot deformity: pes planus, pes valgus, pes planovalgus, pes cavus, and pes transversoplanus

Exclusion criteria:Medical history of knee joint arthroplasty or osteotomyPrevious (surgical) treatment (<12 months) in the ankle, knee, or hip jointX-ray evidence of fixed bone deformity or joint erosionModerate or severe concomitant tibiofemoral OA (Kellgren and Lawrence grade ≥ 3 on anteroposterior radiograph [34])Underlying neurological pathologyKnown underlying rheumatic disease with drug treatmentPrevious treatment with orthopedic FO according to the above concepts while treating the given knee pain indicationAcute muscle/ligament injury (<4 weeks) with associated restriction of the musculoskeletal system

### 2.3. Procedure

After anamnesis and diagnosis by the physician, the participant was instructed to attend a plantar pressure measurement appointment with an experienced orthopedic technician. The navicular index (NI) [35,36] was determined (0.22–0.31 = normal; <0.17 = pes planus; >0.35 = pes cavus), and the arch index (AI) [37] was evaluated using a plantar pressure measuring plate (Multisens, go-tec GmbH, Münster, Germany) (<0.20 = high arch, 0.21–0.26 = normal arch, >0.26 = flattened arch). All parameters for medically indicated foot orthosis fitting were determined based on a 2D foot scan and 3D foot impression. The same orthopedic technician was responsible for manufacturing the FOs for the sample. First, the patient’s medical history and the necessary foot and shoe measurements were obtained using German specifications in the list of aids (product group 08) [29]. A wearing time of at least 8 h per day in the FOs was determined. As a termination criterion during the intervention, an increase in subjectively perceived pain by 2 points or more during the intervention period was defined. Additional therapy was not restricted due to ethical reasons but was documented. Seven of the 24 participants underwent additional physiotherapeutic treatment once a week.

### 2.4. Intervention with Foot Orthoses

The FOs were individually adapted to the patient’s pain, foot, and knee conditions, and a 2D digital foot scan and a 3D foam footprint were conducted (see Figure 1 and Table 2).

### 2.5. Knee Pain, Effectiveness, and Comfort Rating

The participants had to complete the Kujala Anterior Knee Pain Scale [38,39] pre- and post-tests. The questionnaire is a valid and reliable measuring instrument [39]. It is particularly suitable for patients complaining of patellofemoral joint or anterior knee pain [40]. Dammerer et al. [39] validated it for patients with patellofemoral instability using the German version. The total score is 100 points for a symptom-free result and the worst score; therefore, a severe limitation is represented by 0 points.

In addition, 11-item VASs were administered to patients during the intervention time [27,41]. After each week, patients were asked to rate their knee pain perception (0 = no pain and 10 = maximum pain) and document the average steps per day within the week according to their individual smartwatch device.

Furthermore, the overall subjective FO effectiveness and level of comfort of the worn FO were rated using an 11-item VAS (0 (least comfort) to 10 (max. comfort)). This approach was adapted from Murley et al. (2010), who used a 150 mm VAS to measure orthosis comfort [42] because the patients were already used to the scale ranging from 0 to 10 for pain rating by VAS. Participants were instructed to document their daily step counts. Daily steps with FOs during treatment were measured using the participant’s individual smartphone and/or smartwatch and assessed within the post-test.

### 2.6. Statistical Analysis

After preliminary testing for normal distribution using the Shapiro–Wilk test and variance homogeneity using Levene’s test, two repeated measures ANOVA were used for Kujala knee pain scores and 12-week VAS (within-subject factor: time of measurements (ToMs); between-subject factors: treatment groups (SMFO, BMFO)). Physiotherapeutic treatment was added as a between-subject factor to control for its influence on knee pain development. The Kujala anterior knee pain scores were normally distributed for both groups, as assessed using the Shapiro–Wilk test (*p* > 0.05). Therefore, sphericity was assumed. The error variances were homogeneous, as assessed using Levene’s test (*p* > 0.05).

Regarding effectiveness and comfort ratings, Levene’s test demonstrated that variance homogeneity was not present in the comfort rating of the BMFO group, and the Shapiro–Wilk test showed that normal distribution was not present in the SMFO group. Therefore, Welch’s *t*-test was used to assess group differences between subjective FO effectiveness and comfort ratings. Potential differences between the treatment and control groups regarding anthropometric characteristics were additionally explored using independent *t*-tests and the Holm–Bonferroni method.

Generally, a *p*-value ≤ 0.05 was chosen as the statistical cut-off point. Calculations and visualizations were performed using SPSS (IBM, version 29, SPSS Inc., Chicago, IL, USA) and JASP (version 0.19.0, JASP Team, Amsterdam, the Netherlands).

## 3. Results

### 3.1. Consort Flow Diagram

This study adheres to CONSORT guidelines. A total of 34 participants were assessed for eligibility owing to expected dropouts. A total of 27 participants were recruited by the physician. One dropout occurred in the control group during the measurement period (see Figure 2).

### 3.2. Perceived Knee Pain

#### 3.2.1. Kujala Knee Pain Score

There was no statistically significant interaction between ToM and treatment groups (F(1, 20) = 0.01, *p* = 0.92) or ToM and physiotherapeutic treatment (F(1, 20) = 0.09, *p* = 0.77), but there was a significant main effect for ToM (F(1, 20) = 7.23, *p* = 0.01, partial η^2^ = 0.27) (see Figure 3 and Table 3). Bonferroni-adjusted post hoc analysis revealed significantly (*p* = 0.01) increased Kujala anterior knee pain scores between pre- and post-test (Diff_post-pre_ = 10.19 ± 3.79) with Cohen’s *d* = 0.71, which represents a medium effect [43].

The descriptive data show that the ToM scores for the intervention group (IG) were 72.40 ± 11.86 (95% CI [66.40, 78.40]) at pre-test and 83.87 ± 12.21 (95% CI [77.69, 90.04]) at post-test, while the control group (CG) scored 70.27 ± 15.07 (95% CI [61.36, 79.18]) at pre-test and 79.18 ± 15.65 (95% CI [69.93, 88.43]) at post-test (see Table 4).

#### 3.2.2. 12-Week Visual Analog Scales (VASs)

Repeated measures ANOVA showed a highly significant main effect for ToM (F(11, 231) = 12.04, *p* < 0.001, partial η^2^ = 0.36), but there was no statistically significant interaction between ToM and treatment groups (F(11, 231) = 1.40, *p* = 0.18) or ToM and physiotherapeutic treatment (F(11, 231) = 1.47, *p* = 0.15) (see Table 5). Descriptive data are shown in Figure 4 and Figure 5.

Regarding anthropometric data, independent t-tests revealed no significant group differences (height: *p* = 0.64; weight: *p* = 0.94; BMI: *p* = 0.72; NI li: *p* = 0.11; NI re: *p* = 0.05; AI li: *p* = 0.37; AI re: *p* = 0.35). The mean effectiveness rating was 1.42 (95% CI [−0.81, 3.64]) higher in the SMFO group, and the mean comfort rating was 0.36 (95% CI [−0.99, 1.70]) higher for the SMFO group. Welch t-tests showed no significant group differences in effectiveness (t(17.93) = 1.33, *p* = 0.20) and comfort ratings (t(15.03) = 0.57, *p* = 0.58) (see Table 6).

Descriptive and interference statistics of effectiveness and comfort, daily steps, and wearing time per day are shown in Table 6.

## 4. Discussion

The results indicate that both treatments significantly reduced perceived pain in patients with PFP (*p* = 0.01) with a medium effect size (*d* = 0.71). The lack of a statistically significant interaction between the FO treatment and ToM suggests that neither FO approach is superior. Both interventions led to significant pain reduction between baseline and follow-up measurements, as well as in the 12-week pain development measured by VAS. SMFO was rated to be more efficient (Mean_Diff_ = 1.42) and slightly more comfortable (Mean_Diff_ = 0.36) than BMFO on a 11-item VAS; however, statistical analysis revealed no significance for either parameter. In both types of FO, a high level of comfort was detected (BMFO: 7.91 ± 1.87; SMFO: 8.27 ± 1.10); however, it is clinically questionable whether high comfort is an appropriate criterion. For example, Vicenzino et al. [44] investigated predictors of FO success and stated that lower overall comfort had a significant univariate relationship with successful treatment outcomes. Since the FO target has a primarily kinetic and kinematic influence on the foot and lower extremity, this could potentially lead to discomfort. However, a lack of comfort for patients often leads to decreased compliance with wearing FOs. With more than an average of 8000 daily steps with FOs, both treatment groups used FOs almost equally frequently. The results of this study are consistent with those of Lewinson et al. [25] and Skou et al. [19], who confirmed a significant reduction in perceived knee pain. Several possible explanatory models exist for these results. According to Almeida et al. [45], FOs improve the alignment of the knees, hips, pelvis, and spine by adjusting the distribution of plantar foot pressure from the initial contact to the mid-stance phase. Consequently, the improved biomechanical coupling may be a plausible explanation. Overall, influencing foot position and load in both types of treatments can always have the opposite effect in terms of a stronger sensation of discomfort or even pain due to altered joint kinematics; this must always be considered by the physician and orthopedic technician. Both FO treatments had the same therapeutic target (improved biomechanical coupling and pain reduction) based on different mechanisms of action. BMFOs primarily target the stabilization of the rearfoot, help improve plantar pressure distribution, and reduce peak forces, whereas the main target of SMFO is earlier activation of the peroneus longus and tibialis posterior muscles by the lateral and medial spots, which stabilize the ankle joint [46]. Therefore, it can be assumed that a targeted improvement in foot position might lead to improved foot kinetics and kinematics. This stabilization and its influence on the other joints of the lower extremity might explain the reduced discomfort in the knee joint due to biomechanical coupling from the foot and knee [47,48].

When interpreting the data, it must be noted that each patient was treated based on their anatomy and the physician’s initial examination. A clear distinction between the two concepts can still be discussed even though both approaches demonstrate clear differences in therapeutic approach [27]. It must be also considered that footwear FOs can be influenced by the type of shoes participants wore, as (im)proper or (un)supportive footwear may influence the impact of FOs. Therefore, the orthopedic technician checked the FOs and footwear for suitability. While this study showed promising results in reducing PFP patients’ pain perception with both SMFO and BMFO, contextual factors such as placebo effect [49], increased body awareness, and psychological changes likely contributed. Patients may have different pain thresholds, and their reporting could be influenced by personal factors, such as mood, stress, or pain tolerance. Engaging in a clinical trial such as this approach might lead patients to become more aware of their health behaviors, such as exercise habits, activity levels, or posture. The increased motivation to follow prescribed treatments or modify behaviors during the intervention period to positively influence outcomes cannot be controlled. Furthermore, natural and autonomous healing processes of the human body might influence pain relief [27]. However, to measure this effect, another group without any treatment would have been necessary, which was not possible for ethical reasons. Based on the statistical results, SMFO seems to be an effective treatment option compared with the alternative of BMFO. A review by Barton et al. [24] further indicated that a combination of physiotherapy and prefabricated FOs could be more effective than FOs alone in managing PFPS symptoms. Seven of the 24 participants underwent additional physiotherapeutic treatment once a week. In this trial, physiotherapeutic treatment was shown not to have any significant influence on pain development in Kujala anterior knee pain score and 12-week VAS, but only 7 of 24 participants were documented who had physiotherapeutic treatment once a week. The small number of participants and the short treatment period may have played a significant role; the no control group only used physiotherapy. The focus of this approach was on the effect of FOs rather than on further therapeutic methods; however, this should be investigated in future studies.

The sample size can be seen as an average compared with current studies in the field of SMFO effects, such as [30,31,46,50,51,52,53,54]. PFP patients were chosen because of their expected altered gait kinematics [17] and their clinical need for treatment with individually manufactured FOs recommended by a physician. The authors observed a significant improvement in the study’s methodology, as all participants’ FOs were manufactured by the same orthopedic technician with decades of experience. The NI of the sample was reduced [35]. A low NI can favor increased pronation when walking and running and, for this reason, can also be a possible risk factor for PFP. It must be considered that the SMFO group had a lower NI than the control group at baseline measurement. The body mass index of the sample was within the normal range, and the age of the sample corresponded to the target group of younger adults, which was comparable to that reported by Kerkhoff et al. [30]. During participant acquisition, the same physician was responsible for detecting whether the patient fulfilled the defined inclusion criteria and further analyzed foot posture and functionally related lower extremity causes of knee pathology.

Medical examinations included standard diagnostic tools and examinations for physicians and patients [5]. However, the diagnosis of PFP involves different symptoms and manifestations, and it is ultimately impossible to definitively determine whether functional causes of the disease can be found in movement, such as altered tibiofemoral or patellofemoral mechanics [55]. Generally, it must be mentioned that there are several other factors influencing joint disorders, such as hormonal and metabolic [56] or genetic causes [57] that were not controlled. The Kujala anterior knee pain score is a validated measurement tool for detecting subjective pain in patients with knee pain. However, this represents a subjective parameter. The subjective nature of pain reporting can introduce variability into the results. To strengthen the validity by controlling for further disturbing variables, steps and activity levels were assessed, but these were based on self-documentation. It must be considered that there was no standardization in measurement systems (e.g., smartphones or smartwatches), possibly for legal reasons (data law).

To the best of the authors’ knowledge, there was no study that investigated the effects of SMFO on patellofemoral knee patients. This study represents a medically well-supported, strongly controlled methodology supported by an experienced physician and orthopedic technician in a blinded, randomized two-group design. The FOs were made by the same orthopedic technician with over four decades of professional experience. Each FO was individually fitted to a shoe prior to testing. Overall, there is still a lack of randomized controlled clinical trials, and this approach further adds valuable data to the research field. A major limitation, which is why studies on custom-made FOs in general and SMFO in particular are very limited, is that custom-made FO must always be individually adapted to the anatomical and physiological conditions of the patient. This makes standardization of the treatment and, thus, comparability between participants difficult [28]. However, this study examined pain development (ToM) as a within-subject factor. Furthermore, there was an unequal distribution between sexes, including 9 men and 17 women. In addition, investigating the long-term effects of FOs beyond the 3-month intervention period is crucial, as sustained efficacy remains unclear. Incorporating functional assessments and quality-of-life metrics could provide a more comprehensive understanding of the overall impact of FOs on patients’ daily lives.

## 5. Conclusions

In conclusion, the data of this study support the hypothesis that SMFO and BMFO are equally effective treatment options for PFP patients with foot deformities. Although neither approach proved significantly superior, both interventions contributed to meaningful pain relief as the primary therapeutic target. SMFO was rated as more effective and slightly more comfortable by the participants in the follow-up measurement, but no significant group differences were detected. Future studies should investigate kinetic and kinematic changes achieved through FOs, more specifically in all planes, and the differentiation of ankle movements. In addition, larger and gender-equal samples and older age groups should be investigated. Furthermore, more research must be conducted regarding different indications for FO treatment, as there is still no consensus in science regarding when and to what extent FOs can be used for the orthopedic treatment of different lower extremity pathologies. There is a need for further clinical randomized controlled trials and longitudinal studies investigating not only the short term but also the long term.

## Figures and Tables

**Figure 1 biomedicines-13-00038-f001:**
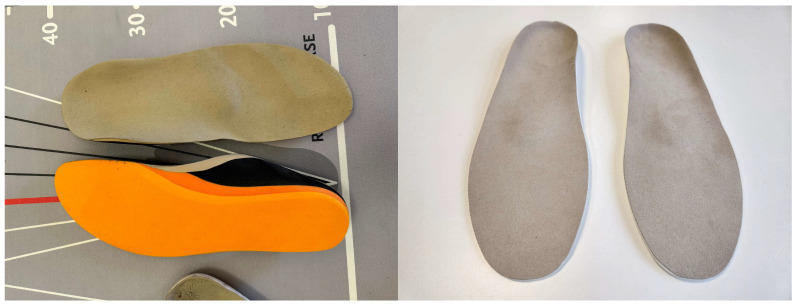
Custom-made sensorimotor foot orthoses (SMFOs) in sandwich material and biomechanical foot orthoses (BMFOs).

**Figure 2 biomedicines-13-00038-f002:**
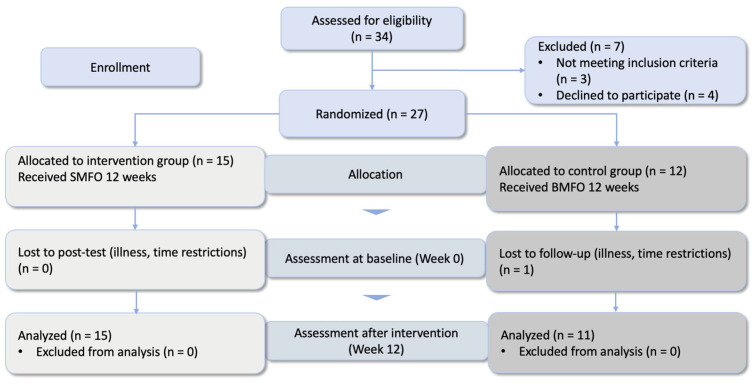
Consort flow diagram of the randomized controlled trial.

**Figure 3 biomedicines-13-00038-f003:**
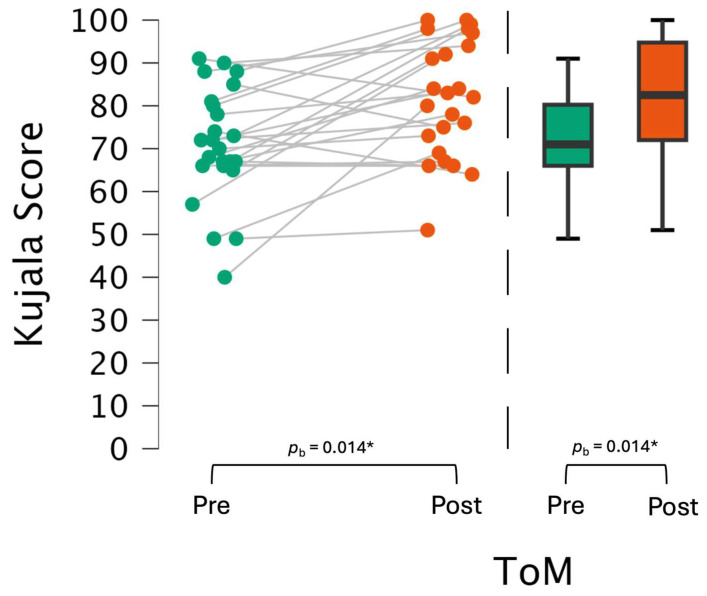
Line diagram and boxplots representing baseline (green dots and bars) and follow-up measurement (red dots and bars). The overall score is 100 points for a maximum good result. Colors: green = pre-test, orange = post-test. * = *p* < 0.05.

**Figure 4 biomedicines-13-00038-f004:**
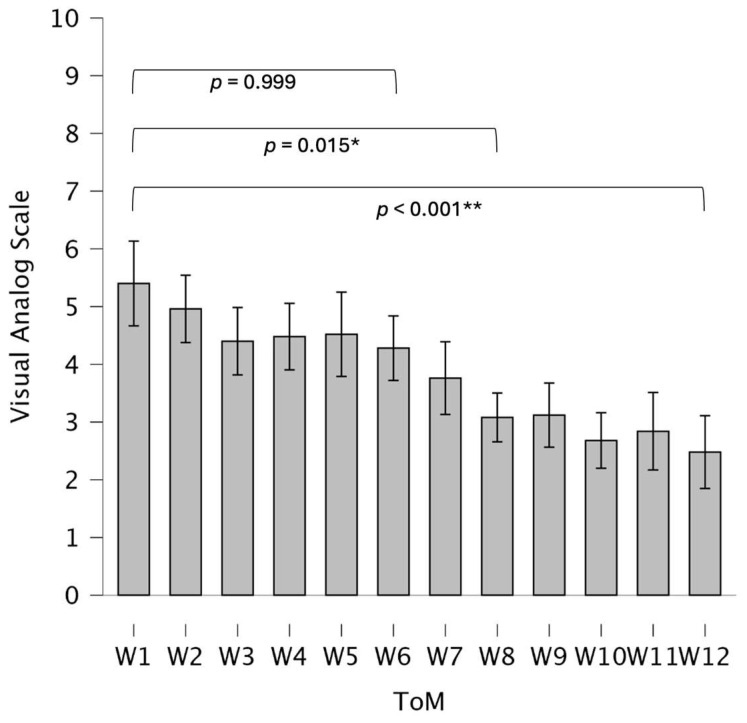
Bar charts representing mean values and 95% confidence interval of VAS development during the intervention period of the sample. *x*-axis = intervention weeks; *y*-axis = VAS score. Abbreviations: ToM = time of measurement; W1–12 = weeks 1–12. * = *p* < 0.05; ** = *p* < 0.01.

**Figure 5 biomedicines-13-00038-f005:**
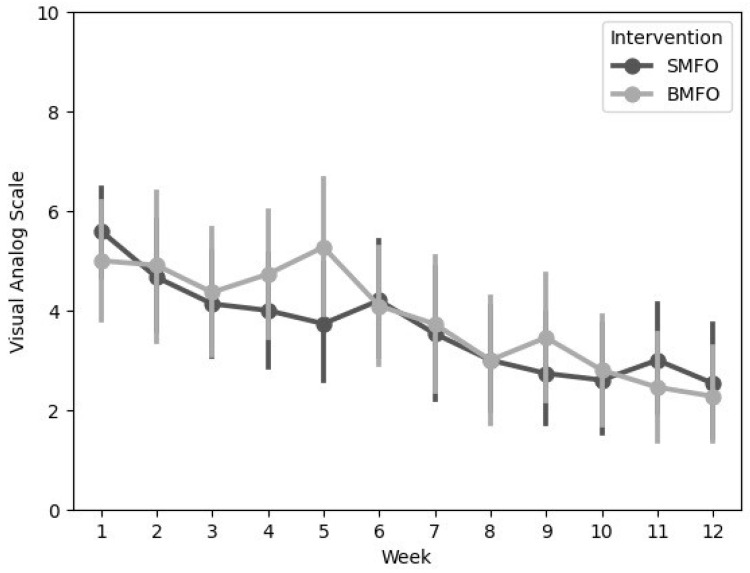
Descriptive plot representing mean values and 95% confidence interval of VAS development in both groups. *x*-axis = intervention weeks; *y*-axis = VAS score. Abbreviations: SMFO = intervention group; BMFO = control group.

**Table 1 biomedicines-13-00038-t001:** Anthropometric data of the sample (n = 26).

		Age (y)	Height (m)	Weight (kg)	BMI	NI Links	NI Rechts	AI Links	AI Rechts
SMFO	Mean	27.27	1.76	75.40	24.30	0.17	0.17	0.22	0.22
	SD	9.19	0.10	18.24	4.64	0.04	0.04	0.07	0.06
	Max	42.00	1.92	115.00	33.24	0.25	0.24	0.31	0.31
	Min	15.00	1.56	47.00	17.26	0.11	0.11	0.09	0.09
BMFO	Mean	29.67	1.75	77.58	25.32	0.20	0.21	0.24	0.24
	SD	13.39	0.09	20.22	6.53	0.06	0.06	0.06	0.02
	Max	54	1.97	120	42.52	0.33	0.33	0.30	0.27
	Min	16	1.64	50	16.14	0.13	0.13	0.10	0.21

Abbreviations: y = years; m = meters; kg = kilograms; NI = navicular index; AI = arch index.

**Table 2 biomedicines-13-00038-t002:** Characteristics and technical data of foot orthoses (SMFOs = sensorimotor foot orthoses; BMFOs = biomechanical foot orthoses).

FO Type	Manufacturer	Primary Medical Target	Elements	Materials
SMFO	Springer Aktiv AG, Berlin, Germany	Stimulating M. tibialis posterior and M. peroneus longus and brevis Stretching plantar fascia and toes	Medial spot (oriented toward M. tibialis posterior tendon at sustentaculum tali) Lateral spot (oriented toward M. peroneus longus and brevis tendon near os cuboideum) Retrocapital bar (supporting the transversal arch and stretching plantar fascia) Toe bar (placing and stretching of toes)	EVA material; sandwich construction consisting of 35 Shore (outsole), 25 Shore (midsole), 35 Shore (top layer)
BMFO	Hema Orthopädische Systeme GmbH, Sömmerda, Germany	Medial arch support Transversal arch support Pressure relief	Heel pad Supination wedge Metatarsal pad (pelotte)	Injection molded foam 25 Shore

**Table 3 biomedicines-13-00038-t003:** Overview of mixed analysis of variance results. Abbreviations: ToM = time of measurements; FO = foot orthoses; physio = physiotherapeutic treatment. x = statistical interaction between variables; * = *p* < 0.05.

	F	η^2^_p_ *p*	*p*
ToM	7.226	0.265	0.014 *
ToM x FO	0.011	5.397 × 10^−4^	0.918
ToM x Physio	0.085	0.004	0.773
ToM x FO x Physio	0.098	0.005	0.757

**Table 4 biomedicines-13-00038-t004:** Descriptive and inference statistics in both groups (intervention group (IG) = SMFO; control group (CG) = BMFO); time of measurements (ToM; pre, post).

ToM	Group	Mean ± SD	CI 95%	rmANOVA
Pre	IG	72.40 ± 11.86	[66.40, 78.40]	Interaction effect: *p* = 0.92 Main effect: *p* = 0.01, η^2^_p_ = 0.27
	CG	70.27 ± 15.07	[61.36, 79.18]	
Post	IG	83.87 ± 12.21	[77.69, 90.04]	
	CG	79.18 ± 15.65	[69.93, 88,43]	

**Table 5 biomedicines-13-00038-t005:** Results of mixed variance of analysis regarding 12-week visual analog scales. X = statistical interaction between variables.

	F	η^2^_p_	*p*
ToM	12.035	0.364	<0.001
ToM x Physio	1.465	0.065	0.146
ToM x FO	1.395	0.062	0.176
ToM x Physio x FO	1.043	0.047	0.410

**Table 6 biomedicines-13-00038-t006:** Descriptive (mean and standard deviation) and inference statistics of subjective effectiveness rating (0 = no help at all; 10 = maximal help) and comfort (0 = no comfort at all; 10 = maximal comfort) by the participants. Average steps with FOs per day and wearing time per day are also shown. Abbreviations: IG = intervention group; CG = control group; FO = foot orthoses.

	Effectiveness Rating	Welch *t*-Test (Effectiveness)	Wearing Comfort Rating	Welch *t*-Test (Comfort)	Daily Steps in FOs	Wearing Time/ Day (h)
IG	7.87 ± 2.23	t(17.93) = 1.33, *p* = 0.20	8.27 ± 1.10	t(15.03) = 0.57 *p* = 0.58	8360 ± 6464	9.68 ± 4.10
CG	6.45 ± 2.94		7.91 ± 1.87		8947 ± 4842	10.00 ± 2.67

## Data Availability

The datasets used and/or analyzed during the current study are available from the corresponding author on reasonable request.

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
