# Peer review of "Pain Reduction in Patellofemoral Knee Patients During 3-Month Intervention with Biomechanical and Sensorimotor Foot Orthoses: A Randomized Controlled Clinical Study"

_biomedicines, 2024, doi:10.3390/biomedicines13010038_

Round 1

Reviewer 1 Report

Comments and Suggestions for Authors

Manuscript has to be improved according to the following suggestions.

Authors should review all references to ensure they cite only sources accessible and readable by the scientific community. For instance, references 28, 29, 30, and 50 are in German and cannot be easily accessed or understood.

References have to be updated to the most recent ones.

Introduction

Lines 69-71: the reference of Collins et al. is lacking. Please add at the end of the sentence.

Line 87: please define the abbreviation SMIs.

Materials and Methods

Please define that BMFO is the control group.

Figure 1 and Table 1 do not include the materials and methods. Please move them to the results section.

Lines 186-187: This belongs in the results section. Please move it accordingly.

Results

This section has to be improved to better visualise the results also adding tables and/or figures. 

Figures 3 and 4: Please include the p-values.

Lines 263-264: The appendix is missing. Please add it.

Lines 260-261: "It would be better to display the results in tables and figures to enhance the comparison. Please include the p-values.

Lines 271-277: Please present this data more clearly, as it is difficult for the reader to follow.

Line 276: p-values are lacking. Please add them.

Table 3: No data on the comparison between the two groups are provided. Please add this along with the specific p-values.

Discussion

Line 290: BMFO: 7.91; SMFO: 8.27 is not clear without also reporting the standard deviation.

Please better report the limits and the strengths of this study.

Lines 380-387: please move this part to the Conclusions section.

References

Some references are quite old.

Reference 23 is not cited in the correct order (after number 22 and before number 24). Please check and correct.

Reference 32 appears to be an abstract. Please replace it with a full-text reference.

Author Response

Dear Reviewer, first and foremost, we want to thank you very much for taking your time to review this manuscript and for this valuable feedback that helped us a lot to further improve our study presentation.
Please find the detailed responses in the attached file and the corresponding corrections in track changes in the re-submitted file.

Reviewer 2 Report

Comments and Suggestions for Authors

The authors reported a randomized clinical trial about Pain Reduction in Patellofemoral Knee Patients during Three- 2 Month Intervention with Biomechanical and Sensorimotor Foot 3 Orthoses. However, they did not find any statistically significant different between biomechanical foot orthoses (BMFO) and sensorimotor foot orthoses (SMFO), they found differences in pain scores and duration of treatment. The authors successfully presented they findings though their well-written manuscript. 

There is no major concerns and I have two minor comments:

1. I recommend to the authors to widen the sample size comparison of other reports with theirs [See lines 343 and 344].

2. The authors can improve their scientific overview by adding a statement in the limitation about the lack of genetic background information of the patients. There are many genetic causes leading to skeletal deformities which can be screened in the early ages by the cooperation of Physicians with Genetic specialist and Treatments by Orthoses can play critical roles in such patients' future. 

Author Response

(The authors gave the same response as above.)

Round 2

Reviewer 1 Report

Comments and Suggestions for Authors

I appreciate the efforts of the authors to improve the manuscript. I have still some comments about data reporting in the tables. Please report p-values in the last column of the tables.

To be clearer data should be reported as mean±standard deviation. Please check all the tables and correct.

Please move Table 3 and its description before Table 4.

Table 3: please report the mean and SD as follows: e.g., 72.40± 11.86. Additionally, please include P-values and a brief description in the manuscript to make it clearer.

Author Response

Dear Reviewer,

again, we want to thank you very much for taking your time to review this manuscript and for this valuable and time-intensive feedback. We are convinced that your comments helped a lot to improve our manuscript.

We have implemented the missing comments and hope that the manuscript now meets the requirements and expectations of the reviewer and journal editor.
